# Novel Characterization Techniques for Multifunctional Plasmonic–Magnetic Nanoparticles in Biomedical Applications

**DOI:** 10.3390/nano13222929

**Published:** 2023-11-11

**Authors:** Rodrigo Calvo, Isabel Rodriguez Mariblanca, Valerio Pini, Monica Dias, Virginia Cebrian, Andreas Thon, Asis Saad, Antonio Salvador-Matar, Óscar Ahumada, Miguel Manso Silván, Aaron E. Saunders, Wentao Wang, Adonis Stassinopoulos

**Affiliations:** 1Mecwins S.A., Tres Cantos, 28760 Madrid, Spain; 2Departamento de Física Aplicada, Universidad Autónoma de Madrid, Campus de Cantoblanco, 28049 Madrid, Spain; 3nanoComposix™, San Diego, CA 92111, USA; asaunders@fortislife.com; 4QuidelOrtho™, San Diego, CA 92121, USAadonis.stassinopoulos@quidelortho.com (A.S.)

**Keywords:** nanoscale magnetism, plasmonic, multifunctional nanoparticle, electron microscopy, SQUID, dark field

## Abstract

In the rapidly emerging field of biomedical applications, multifunctional nanoparticles, especially those containing magnetic and plasmonic components, have gained significant attention due to their combined properties. These hybrid systems, often composed of iron oxide and gold, provide both magnetic and optical functionalities and offer promising avenues for applications in multimodal bioimaging, hyperthermal therapies, and magnetically driven selective delivery. This paper focuses on the implementation of advanced characterization methods, comparing statistical analyses of individual multifunctional particle properties with macroscopic properties as a way of fine-tuning synthetic methodologies for their fabrication methods. Special emphasis is placed on the size-dependent properties, biocompatibility, and challenges that can arise from this versatile nanometric system. In order to ensure the quality and applicability of these particles, various novel methods for characterizing the magnetic gold particles, including the analysis of their morphology, optical response, and magnetic response, are also discussed, with the overall goal of optimizing the fabrication of this complex system and thus enhancing its potential as a preferred diagnostic agent.

## 1. Introduction

The early diagnosis and treatment of diseases such as cancer is a major global health concern. This need has driven the development of innovative techniques that provide both effective and minimally invasive diagnosis and treatment [1,2]. In response, multimodal imaging techniques, which combine different imaging modalities, have been introduced in recent years [3]. Multifunctional nanoparticles are complex nanometrical systems composed of different metals or metal oxides, characterized by their multifaceted nature, that have emerged as interesting tools in this field [4,5]. Unlike single-technology systems that excel at one specific function, multifunctional nanoparticles have diverse capabilities. Their composite nature allows them to take advantage of each constituent material, offering combined properties that a single material does not possess. Single-technology systems are limited by their inherent properties and often require external modifications or additions to impart additional functionalities, making them difficult to manufacture and reducing their effectiveness. Multifunctional nanoparticles, with their inherent diversity of properties, can simplify procedures by reducing the number of agents to be administered or applied, making them a more efficient solution. By exploiting the multiple properties of these particles, they can act as dual contrast agents, overcoming the technical limitations of individual methods and acting as both diagnostic and treatment tools [6].

Plasmonic nanoparticles (NPs) are particles known to exhibit localized surface plasmon resonance (LSPR) [7], a phenomenon in which conducting electrons resonate with an external electromagnetic field, causing the absorption and scattering of specific wavelengths of light. In biomedical applications, gold is the preferred metal for creating these nanoparticles due to its lower toxicity compared to other plasmonic materials such as silver [8,9]. Acting as light antennas, gold nanoparticles (GNPs) facilitate the enhancement of the Raman signal of nearby molecules, known as surface-enhanced Raman scattering (SERS) [10,11]. Consequently, GNPs, in combination with Raman reporters, have been widely investigated for SERS bioimaging in vitro and in vivo treatments [12]. Clinically relevant imaging techniques have used GNPs in photoacoustic [13], SERS, and computed tomography (CT) imaging [14,15,16].

Conversely, magnetic iron oxide (Fe_3_O_2_) or magnetite NPs have been highly valued as MRI contrast agents [17]. Their ability to be manipulated by external magnetic fields allows for targeted applications such as magnetic targeting of cells or tissues [18,19].

The fusion of iron oxide and gold properties led to the creation of multifunctional magnetic gold nanoparticles. These have opened the door to innovative applications that simultaneously utilize MRI and SERS for fast operation and deep tissue penetration [20]. One of the most important applications is the use of multifunctional nanoparticles as nanoheaters activated by an external magnetic field, a technique called hyperthermia. This method involves raising the temperature of a lesion site above 45 °C to induce cell death, an approach that has been shown to be effective in the treatment of cancer [21]. In addition, the dual functionality supports light-driven and magnetically targeted drug delivery [22]. By utilizing both the photothermal response of gold and the magnetic component, this approach improves cancer treatments in terms of specificity and selectivity.

The synthesis of multimodal particles, especially multifunctional magnetic gold nanoparticles (mGNPs), requires careful control of various parameters. Although there are numerous methods for synthesizing these particles [23], much of the existing literature focuses on the preparation of core–shell hybrid nanoparticles (NPs), which consist of a magnetic core surrounded by an Au-based plasmonic shell. Typically, the magnetic core is coated with a layer that acts as a chelating medium, allowing the gold layer to adhere around it. In many cases, this chelating medium is silicon dioxide (SiO_2_) [24,25].

Fabricating these particles is a complex task requiring multiple synthesis steps. Even small changes in the synthesis process can result in variations in the size and composition of the core, as well as the size and properties of the plasmonic shell [26,27,28]. These variations can be experienced both lot to lot but also demonstrated themselves as a source of lot inhomogeneity. Despite advances in mGNP synthesis, achieving consistent size and shape uniformity remains a formidable challenge [24,25]. These variations can lead to unexpected spectral responses or undesirable magnetic properties. Requirements for maintaining strong magnetic responses, forming monodisperse magnetic crystals, ensuring homogeneous layers, and developing silica layers while maintaining tight control of the populations’ particle size all contribute to this complexity.

The incorporation of quality control tools [29,30] to assess particle properties prior to biomedical application is essential. Such tools not only ensure quality and consistency but also provide valuable insights for improving and fine-tuning the synthesis of mGNPs.

In this study, we provide a comprehensive overview of the synthesis of mGNPs and their potential use in biological assays. We also present several innovative characterization techniques for the routine analysis of mGNPs, focusing on aspects such as morphology, chemical composition, optical properties, and magnetic behavior. By elucidating the relationship between synthesis parameters and properties, we aim to advance the optimization of these versatile nanoparticles for a wide range of biomedical applications.

## 2. Experimental Section

### 2.1. Materials and Synthesis

In this study, we examine three lots of mGNPs to be subjected to different characterization processes. These custom nanoparticles, manufactured by nanoComposix, have a structure consisting of a magnetite core (Fe_3_O_4_) surrounded by a silica (SiO_2_) layer and capped with a gold shell. The entire mGNP synthesis process is shown in Figure 1.

Magnetite is a widely preferred material for the synthesis of nanoparticles, not only because of its superparamagnetic properties and high magnetic saturation but also because of its biocompatibility. This makes it an ideal core for multifunctional particles used in diagnostics and characterization. There are several methods to prepare iron oxide nanoparticles, such as magnetite. In this study, we used a synthesis method in which small iron oxide particles (~5–10 nm in size) fuse together to form a larger polycrystalline core.

Iron oxide cores were fabricated by high-temperature reduction of an iron precursor in diethylene glycol in the presence of a polyacrylic acid stabilizing polymer, with the reduction triggered by the rapid addition of base (NaOH). The size of the magnetite core can be fine-tuned by modulating the ratios of the iron, polymer, and base reagents, along with variation of reaction conditions prior to the iron reduction, enabling the production of cores between 40 and 200 nm in size that remain superparamagnetic [31]. Iron oxide core batches were purified from the starting reaction mixture by multiple rounds of centrifugation.

After core synthesis and purification, an amorphous silica layer was introduced. This layer acts as substrate for the attachment of small gold seeds and the subsequent seeded growth of a conformal gold shell. This layer is grown using a modified Stöber process [26,32], which involves the hydrolysis and condensation of TEOS (tetraethyl orthosilicate) in an alcohol-based solution with water, a catalyst (such as ammonium hydroxide NH_4_OH), and the pre-synthesized magnetite cores. By varying the ratio of the TEOS precursor to the magnetite cores, along with the total particle concentration, amount of catalyst, and reaction time, the thickness of the silica layer can be precisely controlled [33]. The silica surface was then amine-functionalized by reaction with APTES ((3-Aminopropyl)triethoxysilane), and the particles were treated with an aluminum salt to improve the stability of the silica surface during subsequent processing steps [34]. The sizes of the magnetite/silica core/shell particles used for Lots 1, 2, and 3 have nominal values of 90 nm, 85 nm, and 93 nm, respectively.

The gold shell was synthesized using the seed-mediated growth method [35]. In the gold seeding phase, prefabricated ~2–5 nm gold THPC-capped seeds (tetrakis(hydroxymethyl) phosphonium chloride) [36] were deposited around the aminated silica-coated magnetic core. These seeds are critical as nucleation sites for the subsequent gold shell deposition. The presence of free gold seeds during the gold nanoshell growth step leads to the homogenous nucleation of free gold particles, and excess THPC-capped seeds may be separated from the cores via centrifugation or countercurrent washing. After seed removal and particle washing, the final gold shell is applied. During this phase, gold is deposited via the reduction of a gold precursor solution (a mixture of gold (III) chloride and potassium carbonate) using formaldehyde. The gold ions reduce and nucleate onto the small gold seeds attached to the silica-coated magnetite core, increasing in size and coalescing to form a complete gold shell. The thickness of the shell can be modulated by adjusting the ratio between the gold precursor and the number of core particles [37,38]. Following completion of the gold shell deposition, a stabilizing ligand is attached to provide steric or electrostatic stabilization of the particles; here, a heterobifunctional thiol-PEG-carboxylic acid with a nominal molecular weight of 5 kDa was introduced, binding to the gold shell through the thiol. The nanoshells were purified from excess reagents and reaction byproducts by repeated rounds of centrifugation and finally dispersed into water. The nominal values of the gold shell thicknesses for Lots 1, 2, and 3 were 26 nm, 23 nm, and 23 nm, respectively. Therefore, the total particle diameters were 136 nm, 131 nm, and 131 nm, respectively.

It is important to note that while all nanoparticle lots underwent the same manufacturing process, each was produced in separate runs. Our primary objective in this work is to evaluate the quality of mGNP fabrication using advanced characterization techniques.

### 2.2. Instruments and Methods

In this study, to obtain transmission electron microscopy (TEM) images, 100 keV JEOL JEM1010 (JEOL, Akishima, Japan) equipment has been used. To obtain the high-resolution TEM (HR-TEM) images, the FEI Titan 80-300 TEM (ThermoFisher, Waltham, MA, USA) was used, a high-resolution transmission electron microscope equipped with a field emission gun and spherical aberration corrector for the imaging lens system, which can achieve resolution down to the information limit of less than 100 pm at 300 keV. The FEI Titan 80-300 TEM can also perform compositional mapping and transmission mode EDS spectrometry using electron energy loss spectroscopy (EELS) with a resolution of approximately ~0.7 eV. This tool allows us to obtain both a compositional spectrophotometry of the sample in a given area of individual particles and an elemental map when this method is combined with scanning transmission electron microscopy (STEM). Samples were deposited onto 200 mesh carbon-coated copper grids for TEM, HR-TEM, and EDS inspections.

The optical properties of the mGNPs were characterized using the dark field single-particle spectrophotometry (DF-SPS) technique, which allows the analysis of the plasmonic signal of thousands of single particles. Sample preparation for this technique consists of drop casting at a low concentration (5 μg/mL) onto a standard silicon wafer with 20 nm of native oxide. Magnetic measurements were performed using a superconducting quantum interference device (SQUID). For the current study, we used an MPMS3 SQUID (Quantum Design^®^, San Diego, CA, USA), a high-sensitivity magnetometer with a 7 Tesla magnet and a sensitivity of up to 10^−8^ emu.

Finally, to evaluate the biological characterization of the magnetic nanoparticles, AVAC technology (Mecwins S.A., Madrid, Spain) was used. AVAC is a highly sensitive digital biosensing platform for ultrasensitive multiplexed detection of protein biomarkers. It uses single plasmonic particles as optical tags to detect the presence of biological markers. Specifically, it counts individual plasmonic nanoparticles that bind to a specific surface in the presence of specific protein biomarkers [39,40].

## 3. Results

The synthesis of magnetic gold nanoparticles (mGNPs) is a complex process, requiring control over the dimensions of multiple materials throughout the multistep fabrication process while avoiding colloidal destabilization and accounting for batch-to-batch variability [23,41,42]. It is, therefore, crucial to characterize the magnetic particles thoroughly, as numerous variables in the manufacturing process can result in particles whose properties differ from those expected, thereby altering their optical or magnetic response. Several state-of-the-art techniques have been employed for the comprehensive characterization of mGNPs, addressing various aspects, such as morphology, chemical composition, optical properties, and magnetic behavior. In this section, we discuss these advanced characterization methods and their relevance to the analysis of mGNPs.

### 3.1. Morphological and Compositional Analysis

The morphology of a nanostructure such as mGNPs includes the dimensions and shapes of its components, and even small changes in these morphological features can induce significant changes in their magnetic and plasmonic properties [27,42]. Therefore, it is crucial to visualize the core–shell structure along with the overall shape and size of the nanoparticle. The sample size needs to be large enough to reflect the properties of the total population and also sampled properly to avoid missing inhomogeneity.

Our study began with the analysis of three batches of mGNP particles using a standard transmission electron microscope (TEM) [43]. Transmission electron microscopy uses a high-energy electron beam that penetrates the sample, and the interaction of the transmitted electrons with the nanometric structure produces an image. This technique facilitates the study of the total nanoparticle size and shape and provides a representative average of the nanoparticle batch size distribution. Figure 2a shows a representative TEM image of each characterized batch. Since the main shape of the nanoparticles studied in this paper is that of a sphere, we used spherical shape detection software with a Hough transform algorithm [44] to identify the diameter of the nanoparticles. It should be noted, however, that this method does not take into account agglomerated particles or any shape that deviates from the spherical standard.

Using hundreds of TEM images, we were able to detect thousands of nanoparticles. This large data set allowed us to generate the histograms for each batch shown in Figure 2b. Interestingly, we found that the diameter values were consistent with the manufacturing values, with an average manufacturing coefficient of approximately 12.8%, as detailed in Table 1. The synthesis recipe clearly defines the predominant diameter values, indicating consistency in the manufacturing process, with a minimal deviation from the values specified by the manufacturer.

After a thorough analysis of the initial two batches via TEM, it was observed that the particle size distribution was remarkably consistent, closely aligning with the manufacturer’s specified dimensions. While there was a minor presence of particles skewed towards smaller sizes, this sub-distribution was minimal and can be attributed to either a more compact magnetite and silica core or slight variations in the gold layer thickness. Importantly, these deviations were substantially overshadowed by the dominant size distribution that conformed to expectations. This not only validates the precision and high-quality control standards employed in the manufacturing process but also underscores its efficacy in producing magnetic nanoparticles of exacting dimensions.

The shape of nanoparticles is one of the most critical morphological features, especially in the context of mGNP development. Ideal particles would be perfectly spherical, but this formation is often elusive. Various factors at different stages of development can prevent the formation of perfect spheres, resulting in deviations from the intended shape.

The Hough method of circular shape detection is often used to analyze these particles. This method allows the extraction of both the perimeter (*P*) and the area (*A*) of the particles. The ratio of these two geometric properties provides a clear indication of the circularity of the particles, showing how closely the manufactured nanoparticles resemble a circle; the closer this value is to one, the more spherical the lot is:(1)$$C=4\pi A/P^{2}$$

Overall, these lots of nanoparticles are in great agreement with a circular shape, as can be seen in Table 1, but the inhomogeneous growth of the gold layer can lead to a loss of circularity.

Although standard TEM allowed us to determine the average size and shape of the magnetic gold nanoparticles, its low electron energy (100 keV, as described in the Section 2.2) was insufficient to penetrate the heavy outer gold layer and reveal the core–shell structure [45]. However, this limitation was overcome using high-resolution TEM. HR-TEM allows materials to be observed at the atomic level, which, in the case of magnetic gold nanoparticles, allows the visualization of mGNPs.

The higher energy electrons easily penetrated the outer layer, revealing the magnetite and silica core, as shown in Figure 3.

Using the capabilities of the spherical shape detection software, we were able to determine average values for the core and shell, as shown in Table 1. Our results showed a coefficient of variation of 5% for the core and up to 20% for the outer gold shell. This variation in the gold shell can be attributed to its growth process, whereby 2–5 nm spherical gold particles adhere to the silica surface and act as seeds for further gold growth. As a result, the gold tends to grow thinner in the interstitial spaces and thicker in the parts where the gold seed adheres initially to the silica. The nanoparticle surface exhibits a certain level of inhomogeneity, leading to a small variability in the gold shell thickness measurable in a few nanometers; this minor discrepancy contributes to a nuanced increase in the coefficient of variation for the outer layer size and a subtle decrease in circularity, as detailed in Table 1. Importantly, these morphological nuances have a marginal impact on the plasmonic signal. Due to the overall shape of the nanoparticles closely approximating that of a sphere, their functionality and efficacy as biological sensors remain robust and highly reliable.

When examining the size of the gold shell and core, the central magnetite component is not visible. This is because the gold masks the iron, another heavy metal, so that even high-energy electrons cannot penetrate the iron–gold combination. Energy-dispersive X-ray spectroscopy (EDS) is a technique used for elemental and chemical characterization and mapping of a nanostructure by X-ray scattering. Using EDS [23,46], we were able to create an elemental map and filter the results based on the energy of the scattered X-rays in the spectrum (as shown in the Appendix A). This revealed the primary components of the particle, such as the magnetic core, which were invisible by HR-TEM. This is clearly shown in Figure 3.

Gold, the primary and enveloping layer of the particle, is most visible in the elemental map of the particle (from Lot 3). However, there is a noticeable absence of this element in the center of the particle where the silica–magnetite core is located. The presence of the magnetic material in the center of the core is confirmed by the iron (Fe_3_O_4_), which is brighter on the iron map. A gap can be seen between the shadow formed by the gold map and the brightness of the iron, indicating the presence of silica.

Although oxygen is more difficult to identify due to the presence of impurities and the shielding effect of the other heavier element, it can be confirmed that it is more abundant in areas stoichiometrically corresponding to silica (SiO_2_) and magnetite (Fe_3_O_4_), i.e., the core.

### 3.2. Further Morphological Analysis: Inhomogeneities and Impurities

While comprehensive analyses of the size and shape of the composite magnetic particles demonstrate the synthesis process capability to produce particles of precisely the intended dimensions and within a narrow distribution, it is worth noting the presence of inhomogeneous formations. Thanks to the advanced HR-TEM tool, it was possible to detect and investigate the origin of these slight inhomogeneities, which not only provided an opportunity to further refine this already efficient manufacturing process but also offered valuable insights that could be transferred to similar nanoparticle synthesis methods in future studies.

One reason for amorphous growth synthesis is the lack of gold seeds in certain places during the outer layer synthesis, as shown in Figure 4a, leaving the core exposed. This is due to several factors. Although the seeds initially adhere uniformly to the surface due to the amination of the silica, there may be gaps where the gold particles do not adhere. This can be caused by defects in the surface treatment, irregularities in the chemistry of the gold seeds, or detachment of a seed during the purification or nucleation processes. However, their effect is minimal as they account for less than 5% of the total particles observed. Consequently, while there is some variation in the final plasmonic response, this is compensated for by the rest of the gold shell.

During TEM and HR-TEM analysis, the formation of shapes deviating from the composite spherical particle was observed in all samples: the dimer formations aggregations of two mGNPs joined at some point during the synthesis process. The inherent magnetic moment of the magnetic particles causes them to aggregate and form clusters, with dimers being the most common [47]. Since the preparation of magnetite cores is the first step in the synthesis of mGNPs, they tend to aggregate from the beginning of the production process. Therefore, we found three types of dimers, as shown in Figure 4b.

Type 1 is formed during the co-precipitation process of the magnetite cores before the growth of the silica layer. At this point, two mGNPs join to form a dimer, which is then enveloped first by the silica and then by the outer gold layer. This type has the most unpredictable optical response, as its shape can vary from a nanorod to a peanut, depending on the growth of the silica and the arrangement of the gold seeds around it.

Type 2 dimers form after the addition of silica once the mGNP core has formed. In this case, two fully formed cores join, and the gold seeds are added, encapsulating the cores in gold. These dimers clearly resemble peanuts, but their optical response is indistinguishable from a Type 1 peanut of the same shape.

Finally, Type 3 is the most common type of dimer formation in all types of particle synthesis. In this scenario, the mGNPs bind after they are fully formed. The Type 3 plasmonic response is always the same, as the two particles are always the same size and shape, although more complex than that of a monomer.

While the optical response of dimers can vary significantly from type to type due to their different shapes, they are clearly distinguishable from mGNP monomers due to their variability and can be easily filtered in an optical inspection.

### 3.3. Optical and Spectral Characterization: Dark-Field Single-Particle Spectrophotometry

The design of the fabricated magnetic gold nanoparticles (mGNPs) enables a plasmonic resonance response that is influenced by the size of the dielectric core, the thickness of the gold layer, and the dielectric properties of the core, the layer, and the surrounding medium [48,49]. Nanostructures with a silica core and a gold layer show pronounced scattering signals in the visible and near-infrared (NIR) spectrum; the plasmonic band shows a shift towards shorter wavelengths as the thickness of the gold shell increases. Therefore, the presence of quality control measures to monitor the plasmonic response of these particles is essential.

Characterization of the plasmonic response of such nanoparticles is typically carried out using UV-VIS absorption techniques [50] or Dynamic Light Scattering (DLS) spectroscopy [51], which uses a variable light source to study the emission spectrum of the suspended particles. However, these techniques have a significant drawback. Because the analysis is performed on the entire population of nanoparticles, the ability to study the behavior of individual particles is lost. As a result, defects such as dimers that form during manufacture are averaged within the overall response of the whole sample and are not easy to study or quantify. Conversely, the ability to obtain a pure spectral signature for the population of the main particles is difficult to achieve.

Alternative methods, such as dark-field spectrophotometry [51,52], can study individual particles, but their application is limited by the low statistical performance resulting from the ability to study only a limited number of nanoparticles per measurement. To address this, in this manuscript, we use the DF-SPS [53,54,55] for single particle analysis. This optical technique allows the rapid and straightforward acquisition of individual spectra from thousands of nanoparticles within a batch in a matter of minutes. DF-SPS is an optical technique capable of measuring the individual sizes of thousands of nanoparticles by integrating the light scattered by the sample at each fixed wavelength and sequentially sweeping over the desired range of spectral components. This allows us not only to observe the spectral variation due to different nanoparticle sizes but also to study the different structures present within a lot of mGNPs.

In the analysis shown in Figure 5a, only mGNP monomers were considered, excluding the dimers shown in Figure 4b, due to the ability of the DF-SPS to filter out the scattering signals of detected particles based on factors such as brightness and plasmonic response [53,55]. However, the number of such formations is minimal compared to the signal from the monomers, less than 30% of the total particles detected. With the ability to filter out other types of particles, we can also detect more complex scattering formations formed by depositing particles on a surface and drying them by drop casting (as described in the Section 2.2), such as trimers, clusters, or even other contaminants [53,55].

The plasmonic response of the three different particle batches is similar, mainly due to their identical dielectric properties and comparable dimensions, as shown in Table 1. However, within the same batch, there is some variation between particles in the plasmonic maximum and scattered emission amplitude, mainly due to variations in the thickness of the outer gold layer observed in the HR-TEM analysis: even small variations can affect the plasmonic response of an mGNP. This is supported by the Mie theory [7,56], which allows the scattering signal of the analyzed mGNPs to be calculated analytically using the optical properties of silica and gold available in reference [57]. A mGNP with a fixed core size of 90 nm and a gold shell of 10 nm for a total particle size of 110 nm has a plasmonic maximum at ~650 nm when surrounded by air (refractive index of 1). When the shell thickness is increased to 25 nm, for a total particle size of 125 nm, the maximum shifts to 600 nm. These findings are consistent with the results shown in Figure 5a, where the emission maxima and minima lie within these values, and the intermediate value is 610 nm, consistent with the manufacturing size of these mGNP batches.

This suggests that despite individual variations due to inhomogeneities in the outer gold layer—resulting from the seed-based growth process—the overall optical response of the batches is within expectations, making them suitable for biomedical applications. It is noteworthy that the mean response of all particles is significantly shifted towards blue for Lot 2. Using the Mie theory, it is confirmed that a reduction in core size (Table 1) causes a shift in the plasmonic response towards the blue spectrum of the nanoparticles. For example, a reduction in core size from 95 to 90 nm results in a shift in the plasmonic maximum of up to 10 nm for the same gold shell, confirming the results shown in Figure 4a.

Figure 5b shows a heatmap of the amplitude and maximum scattering plasmonic emission for all detected particles [55], including both monomers and dimers, within each batch of mGNPs. A prominent region of higher population can be observed where particles whose emission matches that of the monomers shown in Figure 5a are concentrated. This suggests that monomers dominate these samples. Particles that deviate from this primary scattering are located outside of this densely populated region. These outliers may be either inhomogeneously layered monomers, as shown in Figure 4a, or dimer configurations that differ from the composite particles discussed in Figure 5a.

The precision of such narrow emission, both in amplitude and wavelength, is a significant advantage when using these particles as diagnostic biosensors. This precision ensures that small changes in their plasmonic response will be detected immediately when this particle’s surroundings change. Inconsistencies during synthesis could lead to a wider scattering emission in brightness and plasmon resonance wavelength. This is because variations in particle shape and size correlate with significant changes in emission amplitude and wavelength. For context, a particle half the size of those in these batches could have an emission amplitude several orders of magnitude smaller.

### 3.4. Magnetic Characterization

The magnetic properties of magnetite are what make this type of particle special and give it properties for biomedical and industrial applications. These magnetic properties are not only inherent to the magnetite itself but are also significantly influenced by the surrounding silica and gold shell, which can attenuate its response to external magnetic fields. It is, therefore, imperative to employ methods to characterize the magnetic behavior after the complete synthesis process [25].

SQUID magnetometry is an extremely sensitive technique for measuring subtle magnetic fields based on superconducting loops, being the gold standard for measuring the magnetic properties of large quantities of mGNPs. Using SQUID magnetometry, researchers can obtain hysteresis curves and determine the magnetic behavior of the nanoparticles, including their superparamagnetic nature and coercivity. This information is essential for optimizing the synthesis of the magnetite core and assessing the effect of the silica coating and gold shell on the overall magnetic properties of the mGNPs.

Figure 6a shows the hysteresis loops, M(H) versus H(T), of the three batches of mGNPs at a temperature of 20 °C, showing clear superparamagnetic behavior, confirmed by the closed nature of the hysteresis cycles and a practically zero coercivity point, with only a minutely observable ferromagnetic behavior characterized by remanence and a coercivity of ≃0.004 T upon further magnification.

A comparative analysis shows that Lot 1 has a saturation value of 47.6 emu/g, slightly lower than the saturation values of Lots 2 and 3, which are 54.8 and 52.3 emu/g, respectively. This may indicate a larger magnetic core in the latter lots. However, if this were the case, their ferromagnetic behavior would diverge, as larger particles are known to be more ferromagnetic due to the increasing size of the single magnetic domain. The observed similarity in behavior between all three lots, therefore, implies that the magnetic cores must be of similar size.

The lower saturation value of Lot 1 can be attributed then to the diamagnetic contribution of the silica–gold outer layer since a greater layer thickness (Table 1) may result in a more shielded magnetic response from the magnetite core. Despite the shielding effect, the average saturation value of these mGNPs remains commendably high across all batches, comparable to bulk magnetite particles, which range between 70 and 80 emu/g [58]. This suggests a strong response to external magnetic fields, an important requirement for biomedical applications.

Figure 6b further illustrates the zero-field-cooled/field-cooled (ZFC/FC) temperature dependence of the three lots. Consistent with previous findings, Lot 1 exhibits the lowest value of magnetic response. At higher temperatures, the three lots exhibit similar ZFC and FC curves. These curves begin to diverge as temperature decreases, with a maximum observed in the ZFC curve at (T_ZFC.max_) 123.4, 129, and 139.4 T for each batch. This behavior is consistent with superparamagnetic properties: particles become deblocked as temperature increases.

The general approach is that the maximum temperature is a function of the average particle size, while the temperature at which the FC and ZFC curves begin to diverge is the blocking temperature of the largest particles. The difference between the T_ZFC_ max and the blocking temperature (T_B_), 206 K, 195 K, and 196 K, respectively, suggests inter-particle interaction in the three lots, as the magnetic core of the mGNPs is composed of different magnetite particles. Then, it should be assumed that Lot 3 contains the largest particles of the three batches, but these results suggest the presence of particles of different sizes, some of them larger than in the other batches. However, as can be seen in the histogram in Figure 2b, Lot 3 is actually the one with the narrowest size distribution. Also, in Table 1, the size of the nanoparticles is not the largest; however, the size of the magnetic core is the largest of the three batches, which could explain this behavior. Nevertheless, the superparamagnetic behavior remains consistent and favorable in all batches.

### 3.5. Biological Characterization

To evaluate the potential of the synthesized mGNPs as biosensors, an in vitro biological assay was performed. This evaluation is essential to consider these nanoparticles as versatile multifunctional components in diagnostic and therapeutic assays. Experiments were performed both in the absence and in the presence of magnets. Specifically, a low-field flat Nd laboratory magnet designed for 96-well racks (Agilent BioTek 96F magnet, Santa Clara, CA, USA) was used to evaluate how the mGNPs behave in a biological assay under a magnetic field.

The efficiency of mGNPs as sensing particles was evaluated using AVAC technology, which provides an indirect method of biomarker detection by using plasmonic nanoparticles as optical indicators of their presence [59].

For our initial evaluation, the nanoparticles were used in an interleukin-6 (IL-6) detection assay. The nanoparticles were functionalized with an IL-6 detection antibody using EDC/NHS chemistry [60,61]. The IL-6 detection antibody binds to carboxyl groups through its amino groups on the surface of mGNPs, which are activated with EDC at 1 mg/mL and NHS at 1.5 mg/mL in a 1:1 ratio at pH 3.8. The assay was performed on a silicon surface coated with a 20 nm oxide on which an IL-6 capture antibody was immobilized by streptavidin–biotin binding. Amino groups are first introduced by surface silanization, and then carbonyl groups are generated by using glutaraldehyde to form a Schiff base with the amino groups while leaving the other aldehyde group unreacted. These free carbonyls are then used to immobilize streptavidin through a separate Schiff base formation. Finally, a biotin-modified IL-6 capture antibody is attached. The magnet was placed on the opposite side of the silicon surface in experiments under a magnetic field.

The procedure involved incubating the antibody-coated mGNPs from Lot 3 (C = 10 × 10^−14^ g/mL) on this surface at different times, 20 min and 60 min, at 37 °C. This was carried out both in the presence and absence of a magnet at different concentrations of IL-6. AVAC technology was used to individually count each mGNP immobilized on the silicon substrate. This was accomplished by classifying monomers based on their brightness and color, which allowed digital counting of individual mGNPs. Figure 7 shows the results of experiments at different concentrations.

The particle counts after 60 min with magnets are significantly higher than without magnets; the external magnetic field helped to accelerate the kinetic capture process. Even an experiment with a kinetic time three times shorter, 20 min, can achieve similar particle counts as the experiment performed without magnets. The presence of magnets accelerates capture kinetics at all concentrations, with particle counts ~1.5 times higher at IL-6 concentrations below 10 pg/mL. This increases the sensitivity of the particles by reducing the time required for their application in a potential in vivo experiment.

In summary, the integration of magnets with these particles provides dual benefits. Not only does it allow their position to be controlled for potential in vivo experiments, facilitating the detection of affected regions within a subject, but it also improves their kinetics, and therefore sensitivity, by increasing the particle population in the targeted area. In addition, the multifunctional nature of these particles paves the way for the delivery of ultra-localized therapies, reducing the impact on surrounding areas [4,62,63].

## 4. Conclusions

In summary, this work has highlighted the importance of evaluating the morphology and compositional analysis of gold shell magnetite nanoparticles. Advanced characterization techniques have been employed, including TEM, HR-TEM, and EDS for morphology and compositional analysis; DF-SPS for optical characterization; and SQUID for magnetic response. These methods have enriched our understanding of the core–shell structure, overall nanoparticle size and shape, optical and plasmonic response, and biocompatibility of the synthesized mGNPs, ensuring the quality and potential applicability in biomedical pursuits. Notably, we are able to detect even minor discrepancies in size and shape in the produced magnetic particles, being able to refine the production of mGNPs.

We have also shown that in vitro assays play a crucial role in determining the efficacy of these particles as biosensors. The magnetic gold nanoparticles investigated in our study not only exhibit optimal optical responses for such assays but also exhibit accelerated biosensing kinetics when influenced by external magnetic fields.

The characterization of these nanoparticles is critical to overcoming the challenges and realizing the full potential of this versatile nanometric system in the rapidly evolving field of biomedical applications. Comprehensive characterization of mGNPs is essential due to the complexity of their synthesis process and the susceptibility of their properties to variations in manufacturing.

## Figures and Tables

**Figure 1 nanomaterials-13-02929-f001:**
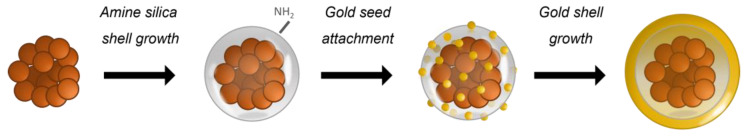
Drawing of magnetic gold nanoparticles synthesis process, from the fabrication of the magnetite (Fe_3_O_4_) core to the creation of the gold shell.

**Figure 2 nanomaterials-13-02929-f002:**
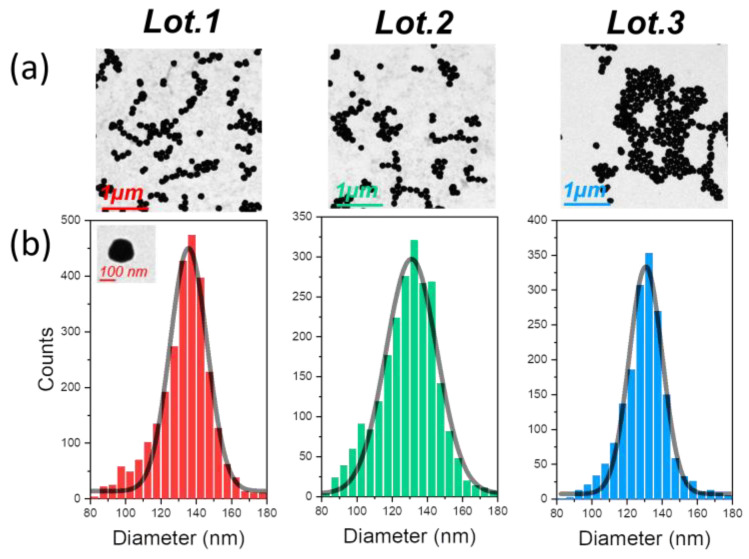
(**a**) Example image of the hundreds of mGNPs found for each batch during TEM inspection from each lot. (**b**) Histogram of the size distribution of the individual particles found by the circular shape recognition software from each lot. The black curve represents the Gaussian fit of the histogram.

**Figure 3 nanomaterials-13-02929-f003:**
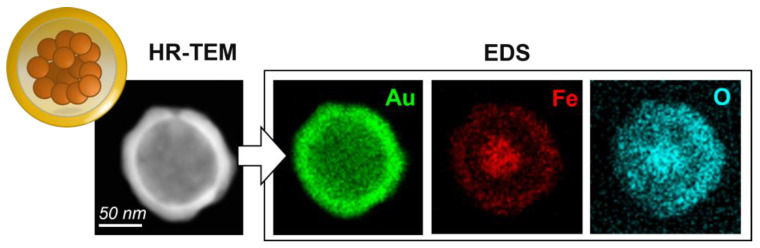
A representative HR-TEM image of a single mGNP from Lot 3, followed by EDS compositional mapping of the exact particle. The gold shell is clearly visible in the HR-TEM image with a brighter white color, while the core appears in a subdued gray color. The following EDS images show the distribution of gold (Au), iron (Fe), and oxygen (O). Each pixel indicates the amount of that element present in that area; the more the element is present, the brighter that pixel is. Because of the way the analysis is carried out through the spectrum (more information can be found in the Appendix A), it is possible to find traces of elements, for example, iron, where they should not be present because of the filter applied to select the specific peak of energies corresponding to that element.

**Figure 4 nanomaterials-13-02929-f004:**
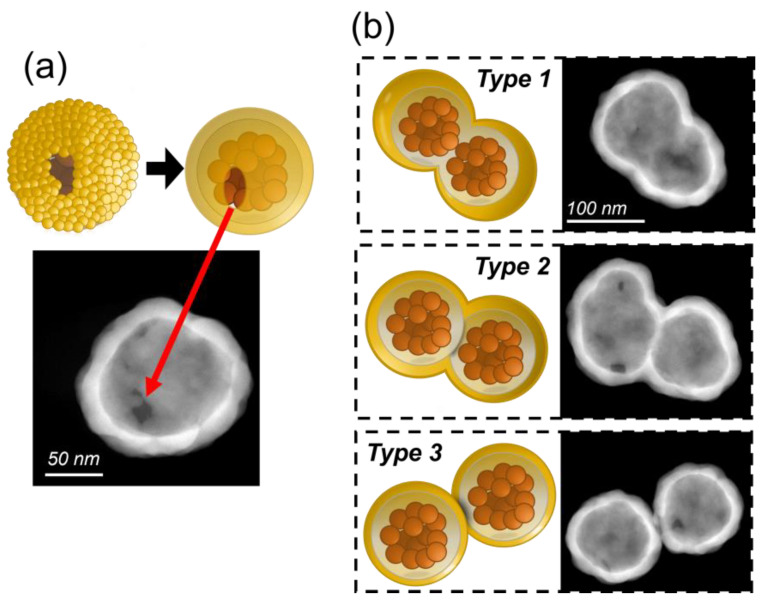
(**a**) Representative HR-TEM image of a single mGNP from Lot 1. This image shows that part of the gold shell is missing due to a missing seed, or several seeds, during the gold shell formation, as depicted in the upper drawing. (**b**) HR-TEM images of the 3 types of dimers found during the synthesis process with an explanatory drawing.

**Figure 5 nanomaterials-13-02929-f005:**
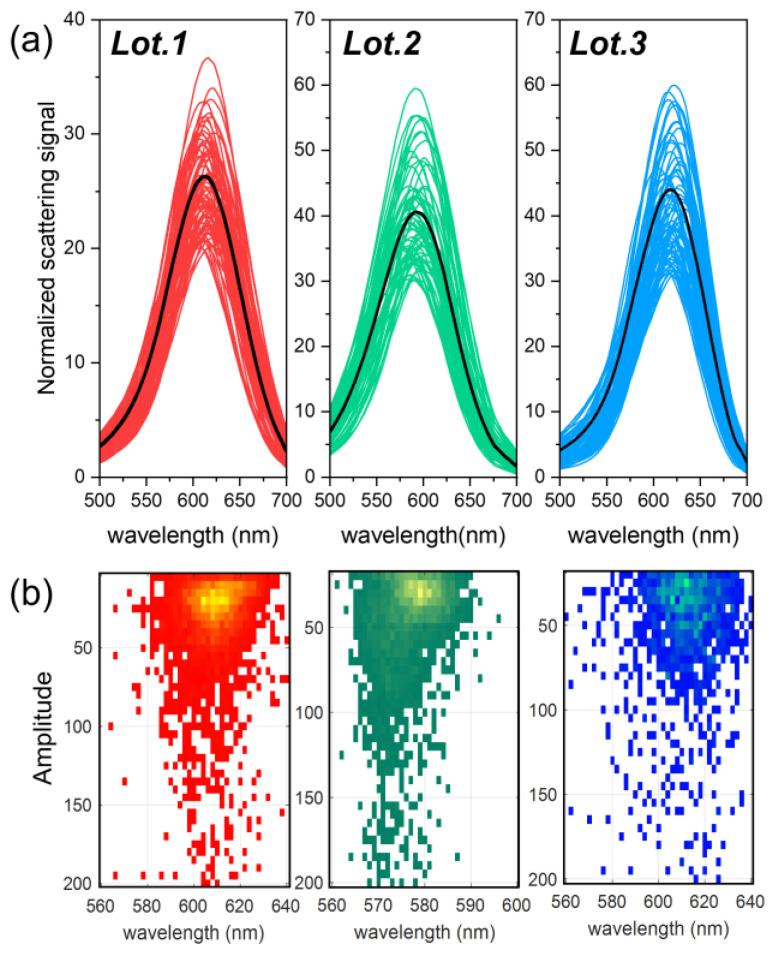
(**a**) Scattering spectra obtained with DF-SPS normalized to background signal, collected from more than hundreds of mGNP monomers; the black curve represents the average of the scattering spectra of the entire monomer population from each batch. (**b**) Heatmap of the scattering amplitude and emission of the plasmonic maximum of ~3000 individual particles from each lot; the lighter the color of the heatmap, the more nanoparticles emit at those spectral values. To calculate both values, each individual emission line has been fitted to a Lorentzian function [55].

**Figure 6 nanomaterials-13-02929-f006:**
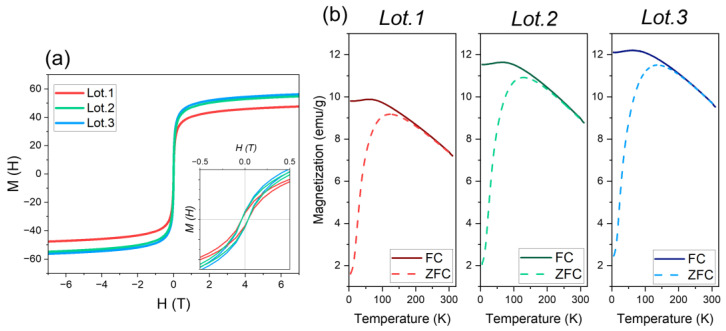
(**a**) Hysteresis loop of the 3 different lots of mGNPs recorded at room temperature. At the bottom of the graph, an inset that shows the loops at lower magnetic fields. (**b**) Variable temperature magnetization curves (ZFC/FC) of each lot of particle collected under 100 Oe field. Solid and dashed lines represent data from field-cooled (FC) and zero-field-cooled (ZFC) measurements, respectively.

**Figure 7 nanomaterials-13-02929-f007:**
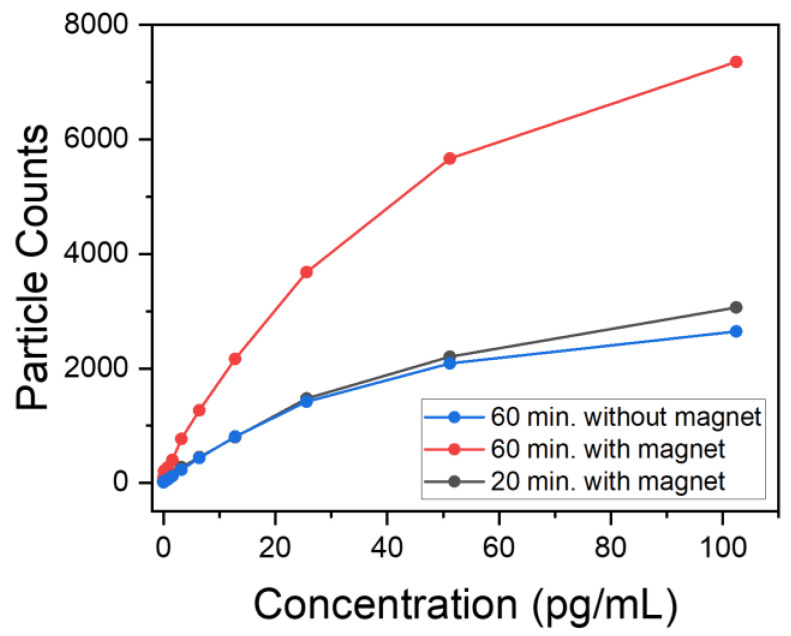
Number of particles detected in the biological assay plotted against the concentration of IL-6 used. The red and gray curves represent experiments performed in the presence of a magnet for two different experimental times, 60 and 20 min, respectively, while the blue curve represents an experiment performed without a magnet for 60 min.

**Table 1 nanomaterials-13-02929-t001:** mGNP sizes of different lots obtained by TEM and HR-TEM.

Lot	TEM Size (nm)	TEM CV %	HR-TEM Core Size (nm)	HR-TEM Shell Size (nm)	Circularity ^1^
Lot 1	136	12.3	95 ± 20	21 ± 8	0.89 ± 0.05
Lot 2	131	14.3	92 ± 21	16 ± 6	0.91 ± 0.05
Lot 3	131	11.9	96 ± 12	18 ± 8	0.91 ± 0.05

^1^ Circularity on column 6 is calculated using Equation (1).

## Data Availability

All details of the data processing and analysis, as well as the results of the HR-TEM, TEM, DF-SPS, and SQUID measurements, are provided in the text and Appendix A.

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
