# Peer review of "Novel Characterization Techniques for Multifunctional Plasmonic–Magnetic Nanoparticles in Biomedical Applications"

_nanomaterials, 2023, doi:10.3390/nano13222929_

Round 1

Reviewer 1 Report

Comments and Suggestions for Authors

The main purpose of this study was to investigate the morphology, component analysis and evaluate the properties of gold shell magnetic nanoparticles, in which several advanced characteristic methods were used. The important part was to clarify the strong relationship between synthesis parameters and performance and to optimize the multifunctional nanoparticles in order to better applications in biomedical region. In addition, TEM, HRTEM, EDS, DF-SPS and SQUID were all used to investigate core-shell structure mGNPs with different sizes and shapes. The characterization of these nanoparticles plays a crucial role in contemporary biomedical fields, enabling this multifunctional nano-system to unleash its full potential and overcome obstacles for future development. In my opinion, the paper is generally well written and most of the content is well described. I felt confident that the versatile nanoparticles the authors synthesized were verified good optical, magnetic response. However, I must state that some descriptions should be improved more clearly. Thus, I have some suggestions in order to make this article more suitable for publication.

1. The advantages of multifunctional nanoparticles in contrast to other single technologies should be carefully described, in the 37 line.

2. The Fe3O4 nanoparticles inside the SiO2 shell should be illustrate clearly in Figure 1.

3. The scale should be labeled in the TEM images of Lot.2 and Lot.3 in Figure 2(a).

4. The caption of Figure 3 should be clearly labeled as an EDS image at the beginning.

5. The spelling word of “wavelenght” should be corrected of Lot.1 in the Figure 5, and it also should indicate (a) and (b) on the figures.

6. The explanation of sensitivity in line 437 should be clarified, “less than” should be changed to “up to”.

Reviewer 2 Report

Comments and Suggestions for Authors

The manuscript by

Rodrigo Calvo et al entitled „ Novel characterization techniques for multifunctional magnetic 2 gold nanoparticles in Biomedical Applications”

is in the scope of a journal Nanomaterials MPDI.

The manuscript reports on the characterization of morphology, composition, and optical and magnetic properties of Fe3O4 nanoparticles surrounded by a silica (SiO2) layer and capped with gold shell nanoparticles produced commercially. Methods of characterization include TEM, EDS, HR-TEM (including statistical analysis of images accounting for size and shape), Dark-Field Single Particle Spectrophotometry, and superconducting quantum interference device (SQUID) magnetometry.

These nanoparticles were tested in the application as biosensors, also in the presence of a magnet. The manuscript is supported by 63 references, with the majority of recent references. The methodology is supported by previously published papers cited in the present work [53-55].

Extensive current literature research on magnetic and plasmonic nanoparticles is provided. The description of the synthesis details is supported by a sufficient number of recent references. The aim of the presented research for testing the synthesis parameters to obtain the tuned optical and magnetic properties of nanoparticles is well justified. The investigated nanoparticles were proved to be a very sensitive biosensor.  

The manuscript is not formally designed according to the conventional requirements.

Section “Experimental” should contain subsections “Materials and Synthesis”, and “Apparatus”. Methods may be presented as separate sections.

Section “Results” should not contain a description of the apparatuses and methodology, but is supposed to present only the results of the research.

The manuscript is a valuable and interesting input in the scientific field of biosensor research and can be recommended for publishing.

Round 2

Reviewer 1 Report

Comments and Suggestions for Authors

I have no further comment 

Author Response

We greatly value the comments and suggestions of reviewers. The feedback provides invaluable guidance to improve the quality and integrity of our work. The reviewer's suggestions ensure that our manuscript meets the high standards set by MDPI Nanomaterials and we are grateful for the valuable guidance provided.